# Effectiveness of interactive voice response-call for life mHealth tool on adherence to anti-retroviral therapy among young people living with HIV: A randomized trial in Uganda

**Agnes Bwanika Naggirinya**[1,2]*, **David B. Meya**[1,2], **Maria Sarah Nabaggala**[2†], **Grace Banturaki**[2], **Agnes Kiragga**[2,3], **Joseph Rujumba**[4], **Peter Waiswa**[5], **Rosalind Parkes-Ratanshi**[2,6]

1 Department of Medicine, College of Health Sciences, Makerere University, Kampala, Uganda, 2 Infectious Diseases Institute, College of Health Sciences, Makerere University, Kampala, Uganda, 3 African Population and Health Research Center, Nairobi, Kenya, 4 Department of Pediatrics and Child Health, College of Health Sciences, Makerere University, Kampala, Uganda, 5 Division of Global Health, School of Public Health, College of Health Sciences, Makerere University, Kampala, Uganda, 6 Department of Psychiatry, University of Cambridge, Cambridge, United Kingdom

† Deceased.
* anaggirinya@idi.co.ug

## Abstract

### Introduction

In people living with HIV, adherence to antiretroviral therapy (ART) is essential for achieving and sustaining viral suppression and reducing mortality. In young people living with HIV, ART adherence and retention remain a challenge with unsatisfactory viral suppression rates despite facility-based intensive adherence counseling that is the standard of care. Few studies have evaluated mHealth adherence interventions among young people living with HIV in resource-limited settings.

### Methods

This was a randomized parallel trial. Young people aged between15-24 living with HIV, initiating ART, or on ART for not more than 6 months at three ART clinics were recruited and randomized to a ratio of 1:1 to either standard of care or the intervention "Call for Life-Interactive Voice Response" with standard of care, between 12th August 2020 and 1st June 2022. The intervention is mobile technology-based software, that has interactive voice response functionalities, with a web-based interface, that allows interaction through the use of voice and tones via a dial pad. The primary outcome was viral suppression defined as HIV RNA below 1,000 copies per mL at 12 months with retention in care and viral suppression at month 6 as a secondary outcome. Descriptive statistics were used for participant characteristics. Primary outcomes and factors associated with viral load failure were assessed using the log-binomial model. All calculations were done using statistical software release 16.0.

**Data Availability Statement:** All relevant data are within the manuscript and its Supporting Information files, 1,2 and 3

**Funding:** ABN received a scholarship from the Infectious Diseases Institute, College of Health Sciences, Makerere University.URL website: https://https://idi.mak.ac.ug/. The sponsors did not participate in any role in the study design, data collection and analysis, decision to publish, or preparation of the manuscript. This was solely by the doctoral student.

**Competing interests:** I have read the journal's policy and the authors of this manuscript have the following competing interests: RPR discloses that Infectious Diseases Institute received funds from Janssen, the Pharmaceutical Companies of Johnson and Johnson for Call for Life Projects and other Project Research.

## Results

A total of 206 participants were recruited and randomized; the mean age was 22.5(SD±1.9) years and 81% (167/206) were female. The intervention had 78.6% (81/103), females, while the standard of care had 83.5% (86/103). Viral suppression at 12 months in the intervention arm was 73.6% (67/91) versus 51.9% (40/77) in the standard of care arm, p=0.01. Retention in care was 88.4% (91/103) in the intervention vs. 74.7% (77/103) in the standard of care arm p=0.01.

## Conclusion

This is the first study in Uganda to demonstrate that mHealth has the potential to improve medication adherence and retention in care among youth living with HIV in Uganda.

## Trial registration

NCT04718974 Registry: clinical Trials.gov.

## Introduction

Globally, 26% of the new HIV infections in 2021 occurred in young people aged 15-24 [1]. This is mainly due to power dynamics and socio-economic and gender-related inequalities. Poor adherence to antiretroviral therapy (ART) is a particular challenge in young people, increasing the risk of ART drug resistance [2] and stalling progress in halting the pandemic. Young people living with HIV (YPLHIV) require special attention due to their inherent risk of poor pill adherence and poor retention in care [3–5].

Sustained, high adherence to ART is essential to achieve HIV viral suppression, leading to improved immunological and clinical outcomes [6, 7] and decreasing the risk of developing ART drug resistance and subsequent transmission of these resistant HIV strains. YPLHIV globally continue to have suboptimal viral load suppression [8], adherence to ART [9], and retention in care [10].

Retention in care is defined as regular attendance at medical scheduled visits, and it offers benefits for both patients and society [11]. For the patient, retention in care ensures access to uninterrupted ART medication, viral load monitoring, and viral suppression, leading to improvement in clinical outcome and survival. For society, retention in HIV care is essential for the effectiveness of HIV treatment through reducing community viral load to prevent HIV transmission [12], as undetectable equals untransmissible [13].

Mobile health (mHealth) involves the use of mobile phones and other wireless technology in medical care [14, 15]. In 2020, over 400,000 young people were newly infected with HIV [16]. As the burden of HIV continues to grow among adolescents and youths, mHealth technology could address many of the healthcare needs of YLWH, including adherence to HIV medications [17, 18], retention in care, and self-management [19], which are critical needs for this priority population.

Few studies have evaluated mHealth adherence interventions among YPLHIV in resource-limited settings [20, 21]. As of March 2021, Uganda had an increase in phone subscribers to 28.3 million from 27.7 million in December 2020 [22, 23], since ownership of a mobile device is common among the youth, the use of mHealth can help increase both engagement and retention in HIV care [24, 25]. The Call for Life-interactive voice response (CFL-IVR) system

utilizes both smart and analog cell phones, despite high phone penetration in Uganda, about 71% of those connected have a basic phone [26], and thus the system has leveraged this to reach out to the young people.

The CFL-IVR system is an automated phone-based technology, it sends out pre-recorded messages with voice prompts, allows participants to provide feedback using their phone key-pads, can schedule calls at particular times, and route calls through any phone network. The system allows real-time data collection and can reach a large and diverse population including those with limited internet and low literacy levels. The system offers individualized pill reminders, clinic visit reminders; health tips messages; and functionality to support self-reported symptoms in English and other languages of preference specific to use cases. The system's development content is in English, however customization of the user interface including translation to other languages like French has been previously done, and any translation to other country-specific languages is possible.

We evaluated an mHealth intervention "Call for life interactive voice response" tool on viral suppression and retention in care among YPLHIV in a randomized trial at a rural district in North-Western Uganda. Adherence to antiretroviral therapy (ART) is crucial for achieving and maintaining viral suppression, which in turn reduces the risk of AIDS progression and mortality [27]. Using the Information - Motivation - Behavioral (IMB) model, we examined the tool's effect on ART adherence behavior and retention in care among young people receiving ART in a rural district in Mid-Western Uganda. The IMB model identifies three key constructs necessary for engaging in a given health behavior: information, motivation, and behavioral skills. Information involved the provision of correct information through the CFL-IVR tool; Motivation through the tool was individualized pill reminder calls, clinic reminders, and symptom reporting and support, to encourage behavioral skills, adherence to daily pills, and clinic visits for drug refills [28].

## Materials and methods

### Study design

Between 20[th] Aug 2020 and 1st Jun 2022, we recruited for a randomized parallel trial in a Kiryandongo district, 225 km away from the capital, which houses the largest refugee settlement in Uganda. The follow-up period started on 27[th] Jan 2021 and was completed on 1[st] Jun 2022.

### Study sites

Recruitment was from three ART clinics run in a health facility II level (Nyakadot Health Centre II), health facility III (Panyadoli Health Centre III), and a hospital (Kiryandongo Hospital).

The health facility level III is located within a refugee settlement and is the busiest clinic in the district. Panyadoli HC, located at the Kiryandongo Refugee Settlement in Bweyale, Uganda, provides healthcare services to over 100,000 refugees and persons in the community.

### Participants

Young people living with HIV 15-24 years, initiating ART or on ART for ≤ 6 months registered at any of the three study sites were eligible for randomization in a ratio 1:1 to either Standard of Care which is usual care, or the intervention "CFL-IVR plus standard of care".

### Study procedures

**Randomization and concealment.** Block randomization at a 1:1 ratio was done to either standard of care or Call for life-IVR with standard of care. Computerized varying blocks of six

randomizations were done by an independent statistician who kept the treatment code list. We followed sequentially numbered, opaque, sealed envelopes; the treatment allocation envelopes (TAE) were tamper-evident, and identical (except the pre-written study number). One envelope was created for each particular trial subject. Each envelope had a study identification number. The envelopes were secured in a lockable drawer kept by an independent non-study staff. Inside each envelope was a piece of white paper printed clearly with the study number and the allocation given either "Call for life" or "standard of care". All participants were given a basic mobile phone after randomization to minimize differences in baseline characteristics and ensure no participants were excluded due to lack of a phone. Blinding was not possible and was not done.

**Standard of care (usual care) arm.** Participants received minimum care facility-based services that occurred 6 monthly and included: Clinical evaluation and monitoring for World Health Organisation (WHO) clinical stage of disease; ART therapy regardless of clinical stage or CD4; nutritional assessment, counseling & support, sexual reproductive health services, screening and management of opportunistic infections e.g. TB, isoniazid prophylaxis if eligible, screen and treatment of co-morbidities, adherence counseling and support, psychosocial support, assessment of stigma, family support, and client disclosure to family [29]. Participants were seen every six months as per the Ministry of Health Consolidated guidelines for the prevention and treatment of HIV [29].

**Intervention description (Call for life-IVR mHealth arm).** The CFL-IVR tool is a software that is based on open-source Mobile Technology for Community Health (MoTeCH). MoTeCH was initially developed by the Grameen Foundation and the University of Southern Maine with the support of Janssen, the Pharmaceutical Companies of Johnson & Johnson [30]. CFL-IVR offered individualized pill reminders, clinic visit reminders; health tips messages; and functionality to support self-reported symptoms [31]. The intervention arm received weekly health information on sexual reproductive health, HIV & ART, positive living, prevention of mother-to-child transmission, and AIDS-related opportunistic infections. The daily pill reminders and 6-monthly clinic reminders were individualized, and participants had an opportunity to report symptoms at the end of a day's pill reminder and when the need arose, in addition, they received standard of care as per Uganda Ministry of Health guidelines for HIV care and management [29].

Demographics, medical history, self-reported stigma, socio-economic evaluation, HIV basic Knowledge, and Sexual Behaviour questionnaire completed.

Viral load testing: In August 2014, Uganda's Ministry of Health launched a centralized Viral Load (VL) testing system at Central Public Health Laboratories (CPHL) to monitor the response of Anti-retroviral therapy (ART) nationally. Samples from facilities involved in HIV care and management are transported and tested at CPHL through the National Laboratory Sample Transport system. Participants' samples are drawn based on eligibility criteria, with six-monthly testing for adolescents and young adults, and the hub system used to transport to CPHL for testing.

**Outcomes.** Viral load (VL) testing was performed at baseline (for those on ART for 6 months), and repeat VL samples were taken at month 6, and month 12 study visits. Participants were seen at months 6, and 12 visits, and all procedures were repeated on these visits. The primary outcome was viral suppression at 12 months, defined as viral copies below 1000/ mL at month 12 visit. The secondary outcomes were retention in care defined as the proportion of participants that turned up for care at month 12, and viral suppression at month 6 visit.

**Statistical analysis.** The sample size calculation was based on the primary outcome using the openEpi.com sample size calculation formula for randomized clinical trials [32]. We used a two-sided significance level (alpha) of 0.05, a power of 80, with a ratio of unexposed: exposed

of 1. The primary endpoint was viral load suppression at month 12, defined as viral load count below detection (<1000 copies/ml); we estimated 79% viral suppression in the control arm based on previous results in the study area [33], with 94% virally suppressed in the intervention arm [34]. Factoring 10% loss to follow-up, the total sample size was estimated at 206 (103 for each arm). The primary endpoint was viral load suppression at 12 months. Data was collected and managed using Research Electronic Data Capture (%) 10.0.25 - © 2023 Vanderbilt University hosted at Infectious Diseases Institute. The primary outcome was estimated using a log-binomial regression model adjusting for facility level and ART status at entry at month 12, and sensitivity analysis of the primary outcome was estimated using a log-binomial regression model adjusting for more factors including facility level, ART status at entry, age, gender, and reasons for not taking pills: Simply forgot. The percentages were predicted from the adjusted model.

Viral load results at month 12 were analyzed, and proportions of those with detectable VL and those with suppressed VL were estimated in both arms; risk ratios and factors associated with viral load failure were calculated with a p-value of 0.05 taken as statistically significant. The secondary outcome (viral suppression at month 6) was estimated using a log-binomial regression model adjusting for facility level and ART status at entry at month. For the secondary outcome on proportions retained in care at 6 and 12-month visits, the difference in proportion and its confidence interval across the groups was obtained using the binomial method (unadjusted).

Per protocol analysis was conducted including only the participants attending the month 12 study visit for assessment of the proportions of viral load suppression. Statistical analysis was performed using Stata version 16.0 (Stata Corp, College Station, Texas, USA).

Data monitoring committee: We had a Data Steering Committee (DSC) consisting of independent members with expertise in relevant clinical specialties, and statistical data management among which the doctoral committee supervisors were. Their responsibility was to ensure that the safety of study subjects was protected while the scientific goals of the study were being met, and generated data queries to the study team.

**Ethical approvals and processes.**   The study was approved and cleared by the School of Medicine Research Ethics Committee (Rec Ref# 2019-083) and Uganda National Council of Science & Technology (UNCST Folio# HS576ES). At baseline study information was given and written informed consent was obtained from all participants 18 years and above. For those below 18 years of age, parental/guardian consent was obtained and assent from the minors. For the emancipated minors (16 -17 years) according to UNCST guidelines, no parental consent was required. As per the standard of care, all participants on ART for 6 months and with unsuppressed VL were offered intense adherence counseling (IAC) by the study research assistants and the nurse counselors in the ART clinic. The participants received monthly counseling sessions for three consecutive months.

Reimbursement was waived since it would interfere with a willingness to return for clinic visits as the majority would be returning for reimbursement fees and this would not give the pragmatic retention rates. Trial registration was done on clinicaltrials.gov registry NCT 04718974.

## Results

We screened 296 participants of which 206 were enrolled and were randomized in a 1:1 ratio to either Call for Life Intervention or SoC. At month 12, only those who returned for follow-up were included in the analysis (91 in the intervention; 77 in the standard of care arm), "Fig 1".

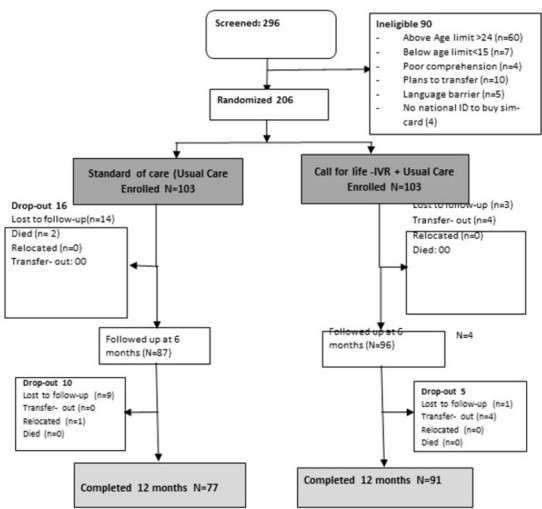

**Fig 1. Consort diagram.** Fig 1 shows the participant flow from recruitment to study exit.

## Baseline characteristics

The participants' mean age was 22.4(2.1) years, 81% (167/206) were females, 88% (183/206) had a sexual partner, and 25% (52/206) had a secondary level of education. About 64% (n=131) of participants were newly initiated on ART and not eligible for viral load testing at baseline. Of 75 young adults that had viral load tests done at baseline (39 Intervention and 36 SoC); 22.3% (46) had undetectable VL, 21 in the intervention arm while 25 in the standard of care arm, the arms did not have major differences at baseline (Table 1).

## Primary and secondary outcome

Viral suppression by arm: At 12 months, 73.6% (67/91) of those on intervention had a suppressed VL compared to 51.9% (40/77) SoC, with a difference in percentage and 95% Confidence Interval of -0.20(-0.35 to -0.06) which was statistically significant P=0.01. Secondary outcomes which included viral suppression at month 6 did not differ statistically between arms (Table 2).

Retention in care at 12- and 6-month visits: There was a statistically significant difference in percentage (95% CI percentage points) for retention in care at month 6 of 8.7% (0.2% to 17.3%) and 12 months, a difference of 13.5% (3.1% to 24.0%) in favor of the intervention (Table 3).

Factors associated with non-suppression using the log-binomial model after adjusting for various variables included, facility level, education level, employment, and those who report the reason for missing pills as "simply forgot". Various factors were assessed for viral load failure and Table 4. shows the unadjusted and adjusted model with factors linked to non-suppression (Table 4).

ART adherence and reasons for missing medication at month 12 bivariate analysis with Pearson chi-square test and Fisher's exact test: Around 34 participants reported ever missing pills, and the reasons for non-adherence leading to missed pills at month 12 visit included; being away from home (n= 30), simply forgot (n=44), not wanting to be seen taking drugs (n=31), ran out drugs (n=32) and feeling healthy (n=10). These reasons differed significantly between arms (Table 5).

**Table 1. Baseline characteristics between study arms.**

| Variable | Interventional N=103 | Standard N=103 |
|---|---|---|
| Age Mean (SD) | 22.5(1.93) | 22.3(2.25) |
| **Gender** | | |
| Male | 22(21.4%) | 17(16.5%) |
| Female | 81(78.6%) | 86(83.5%) |
| **ART status at entry** | | |
| ART naïve | 63(61.2%) | 68(66.0%) |
| On ART >6 months | 40(38.8%) | 35(34.07%) |
| **Baseline VL results* (N/A if initiating ART*) n=75** | | |
| ≥1000 copies/ml | 18(17.5%) | 11(10.7%) |
| <1000 copies/m | 22(21.4%) | 24(23.3%) |
| **Sexual Partner** | | |
| Yes | 94(91.3%) | 89(86.4%) |
| No | 9(8.7%) | 14(13.6%) |
| **Disclosure of status** | | |
| Yes | 95(92.2%) | 94(91.3%) |
| No | 8(7.8%) | 9(8.7%) |
| **Highest level of schooling** | | |
| No formal Education | 2(1.9%) | 4(3.9%) |
| Primary | 69(67.0%) | 79(76.7%) |
| Secondary | 29(28.2%) | 17(16.5%) |
| Technical | 3(2.9%) | 3(2.9%) |

SD=standard deviation

* Only 75 participants had baseline viral load tests at entry

**Table 2. Viral suppression at 12 months, 6 months.**

| Outcome | Intervention (CFL-IVR) | Standard of care (SOC) | Difference (95% CI) Percentage points | P-value |
|---|---|---|---|---|
| **Primary Outcome: Viral suppression at 12 months, per-protocol population a** | | | | |
| Total Participants | 91 | 77 | — | |
| HIV-1 RNA levels, n (%)[1] | | | | |
| <1000 copies/ml | 67(73.6%) | 40(51.9%) | — | |
| ≥1000 copies/ml | 24 (26.4%) | 37 (48.1%) | -0.20 (-0.35 to -0.06) | <0.01 |
| **Sensitivity analysis, HIV-1 RNA levels** | | | | |
| ≥1000 copies/ml, adjusted[2] % | 31.6% | 41.2% | -0.10 (-0.25 to 0.06) | 0.21 |
| **Secondary outcomes: Viral suppression at month 6, complete cases b** | | | | |
| Total Participants, n | 96 | 86 | — | |
| HIV-1 RNA levels, n (%)[3] | | | | |
| <1000 copies/ml | 61(63.5%) | 46(53.5%) | -0.09 (-0.23 to 0.06) | 0.23 |
| ≥1000 copies/ml | 35 (36.5%) | 40 (46.5%) | — | |

[1] The primary outcome was estimated using a log-binomial regression model adjusting for facility level and ART status at entry at month 12.

[2] The sensitivity analysis of the primary outcome was estimated using log-binomial regression model adjusting for more factors including facility level, ART status at entry, age, gender, and reasons for not taking pills: Simply forgot. The percentages were predicted from the adjusted model.

[3] The secondary outcome (viral suppression) was estimated using a log-binomial regression model adjusting for facility level and ART status at entry at month 6

[a]. This analysis only includes participants who had a 12-month visit

[b]. This analysis only includes participants who had a 06-month visit

**Table 3. Retention in care at 12 months, and 6 months visit.**

| Outcome | Intervention (CFL-IVR arm) | Standard of care (SOC arm) | Difference (95% CI) Percentage points | P-value |
|---|---|---|---|---|
| **Secondary outcomes: Retention at 12 months, n (%)[1]** | | | | |
| *Total Participants* | *103* | *103* | | |
| Retained, n (%) | 91 (88.4%) | 77 (74.7%) | 13.5% (3.1% to 24.0%) | 0.01 |
| Not Retained, n %) | 12 (11.7%) | 26 (25.2%) | | |
| **Secondary outcome: Retention at 6 months** | | | | |
| *Total Participants, n* | *103* | *103* | — | |
| Retained, n (%) | 96 (93.2%) | 87 (84.5%) | 8.7% (0.2% to 17.3%) | 0.05 |
| Not Retained, n %) | 7 (6.8%) | 16 (15.5%) | | |

[1] The difference in proportion and its confidence interval across the groups was obtained using the binomial method (unadjusted). [2] The difference in proportion and its confidence interval across the groups was obtained using the binomial method (unadjusted).

Added value of this study: The study introduces an interactive voice response system, which doesn't depend on literacy levels, it is secured with a 4-digit PIN code to ensure confidentiality and privacy. It reaches out to the youth in a most appealing way. This study is the first mHealth technology to show a positive effect on YPLHIV in Uganda and indeed sub-Saharan Africa. The large effect seen in the study of a 21.7% improvement in viral suppression is very encouraging as a tool to support drug adherence and retention in care for young people. By using a simple (analog) phone, this digital intervention can easily be accessed by the majority of YPLHIV in Africa. This tool has pill reminders, clinic appointment reminders, health

**Table 4. Factors associated with viral load failure using log-binomial model.**

| Variables | Unadjusted RR (CI) | p-value | Adjusted RR (CI) | P-value |
|---|---|---|---|---|
| **Arm** | | | | |
| Standard | Ref | | Ref | |
| Intervention | 0.54(0.36-0.83) | 0.01 | 0.72(0.46-1.13) | 0.16 |
| **ART status at entry** | | | | |
| ART naïve | Ref | | Ref | |
| On ART > 6 months | 0.89(0.59-1.36) | 0.60 | 0.94(0.67-1.32) | 0.73 |
| **Facility level** | | | | |
| HC III | Ref | | Ref | |
| HC IV | 1.90(1.09-3.32) | 0.02 | 1.80(1.05-3.06) | 0.03 |
| Hospital | 1.84(0.98-3.46) | 0.06 | 2.31(1.31-4.04) | <0.01 |
| **Education level** | | | | |
| No formal education | Ref | | Ref | |
| Primary | 0.50(0.29-0.85) | 0.01 | 0.42(0.32-0.54) | <0.01 |
| Secondary | 0.47(0.24-0.89) | 0.02 | 0.48(0.33-0.70) | <0.01 |
| Technical/University | 0.40(0.11-1.41) | 0.16 | 0.40(0.13-1.21) | 0.11 |
| **Current working** | | | | |
| Yes | Ref | | Ref | |
| No | 0.55(0.23-1.34) | 0.19 | 0.52(0.36-0.76) | <0.01 |
| **Simply forgot** | | | | |
| No | Ref | | Ref | |
| Yes | 1.84(1.26-2.70) | <0.01 | 1.55(1.12-2.14) | <0.01 |

HC: health center

**Table 5. Reasons for missing ART medication at month 12, based on bivariate analysis.**

| Reason | Interventional | Standard | Total | p-value |
|---|---|---|---|---|
| | (N = 91) | (N = 77) | (N = 168) | |
| **Away from home** | | | | <0.001 |
| Never | 85 (93.4%) | 53 (68.8%) | 138 (82.1%) | |
| Rarely | 5 (5.5%) | 7 (9.1%) | 12 (7.1%) | |
| Sometimes | 1 (1.1%) | 17 (22.1%) | 18 (10.7%) | |
| **Simply forgot** | | | | <0.001 |
| Never | 86 (94.5%) | 36 (46.8%) | 122 (72.6%) | |
| Rarely | 4 (4.4%) | 23 (29.9%) | 27 (16.1%) | |
| Sometimes | 0 (0.0%) | 17 (22.1%) | 17 (10.1%) | |
| Often | 1 (1.1%) | 1 (1.3%) | 2 (1.2%) | |
| **Did not want to be noticed taking pills** | | | | <0.001 |
| Never | 85 (93.4%) | 52 (67.5%) | 137 (81.5%) | |
| Rarely | 2 (2.2%) | 11 (14.3%) | 13 (7.7%) | |
| Sometimes | 3 (3.3%) | 13 (16.9%) | 16 (9.5%) | |
| Often | 1 (1.1%) | 1 (1.3%) | 2 (1.2%) | |
| **Ran out of drugs** | | | | <0.001 |
| Never | 85 (93.4%) | 51 (66.2%) | 136 (81.0%) | |
| Rarely | 5 (5.5%) | 23 (29.9%) | 28 (16.7%) | |
| Sometimes | 0 (0.0%) | 3 (3.9%) | 3 (1.8%) | |
| Often | 1 (1.1%) | 0 (0.0%) | 1 (0.6%) | |
| **Felt healthy** | | | | 0.002 |
| Never | 91 (100.0%) | 67 (87.0%) | 158 (94.0%) | |
| Rarely | 0 (0.0%) | 6 (7.8%) | 6 (3.6%) | |
| Sometimes | 0 (0.0%) | 4 (5.2%) | 4 (2.4%) | |

A bivariate analysis of reasons for not taking pills at 12-month visit used the Pearson Chi-square test and Fisher exact test for assessing association. Commonly reported reasons for missing drugs were "simply forgot" and "run out of drugs" on rare circumstances.

tips, and remote symptom reporting which can be accustomed to an individual's needs and schedule. All patient data can be accessed on the tool dashboard by the clinician who has rights and login credentials.

## Discussion

This is the first study to show improvement in viral suppression and retention in care using mHealth among YPLHIV in a resource-limited setting. The Call for life intervention had a 21% positive effect on viral suppression at the end of the study. Significant viral load suppression proportions of 73.6% in the intervention arm as compared to 51.9% in the standard of care arm at twelve months, and retention rate of 88.4% retention in care for the intervention arm as compared to 74.7% in the standard of care arm. Our population was mainly young females, this is not surprising and concurs with literature and surveys done in Uganda and elsewhere in the world, where the prevalence of HIV is higher among females, 2.9% versus 0.8% [35].

Most new HIV infections among young persons aged 15-24 years, occurring among adolescent girls and young women, are attributed to gender violence, limited access to prevention information, and lack of comprehensive and accurate knowledge about HIV. However, due to

the availability of HIV testing during antenatal visits and child health days, females are captured more.

Retention in care represents the process of ongoing participation in HIV medical care, in our study, the average retention rates in our study were 93.6% for the intervention arm compared to 86.9% in the standard of care arm, a non-significant difference of 6.7%. At 6 and 12-months clinical follow-up, retention rates in intervention versus standard of care were 93.2% and 84.5%; 88.4% and 74.7% respectively, with the latter having a statistically significant difference of 13.5% between arms in favor of the CFL-IVR intervention. All our retention rates were higher than retention rates reported by previous mHealth in HIV studies; WelTel Retain trial that used interactive text-messaging as an intervention [36], a retrospective study among a similar population of young people [37], and a cross-sectional study [5] with similar settings although the latter two never used mHealth intervention. These mHealth studies used short-text messaging services which might not be appealing to the youth compared to interactive voice response intervention. Further, the IVR doesn't depend on or require the beneficiary to be literate.

This study is one of the first to demonstrate that a mobile health intervention can have a positive impact on retention in care among young people, one other mHealth intervention in an adult population, had a similar impact but the population was predominantly male (63%) and in a developed setting [38]. A systematic review of mHealth trials reported over half of the studies had a significant positive effect on primary outcomes of retention and adherence [39]. A six-month pilot study in the United States used text messaging and reported similar retention rates [38]. Globally, studies have reported challenges of retention in care among youths living with HIV [40–42] and the world is still lagging on targets set for young people [1]. More female participants were lost to follow-up during the study. This attrition was highest in the standard of care arm. This is similar to a cross-sectional study in southwestern Uganda, where being male was associated with retention in HIV care [5]. More studies have reported males being lost to follow-up contrary to our findings [43–45]. The barriers to retention in care were mainly fear of disclosure to others as previously reported in a retrospective study of a mixed population [46], social isolation [47], stigma [48, 49], and unfriendly clinic settings and services. There are still noteworthy gaps in the HIV clinical care cascade among the youth as we pursue the Joint United Nations Programme on HIV and AIDS 95–95–95 targets. Achieving these targets could potentially be attained through the use of mHealth tool kits [48, 50].

ART adherence and viral suppression: Adherence to ART is vital for viral suppression, various mHealth interventions have been deployed in the adult population [51], however, the adolescents and young people living with HIV still lack interventions. In our study, the mHealth Call for Life-IVR tool had a significant impact on viral suppression, our finding is similar to one study in Cameroon, although the population here had TB coinfection [52]. In the United States, mHealth had a positive impact, among cisgender MSM and transgender women [53].

Our study is among the first mHealth studies using IVR in young people living with HIV in the sub-Saharan region that has shown a significant effect on viral suppression. Most mHealth interventions have been tested in studies with small sample sizes to assess acceptability, and non-HIV conditions [54], and the majority have used short-message services (SMS) [18, 55], whereas our intervention was interactive voice response. Other studies have found no association with viral suppression but only with medication adherence [21, 56].

Due to confidentiality, the ability to personalize reminders, and the option of call-in for symptom reporting, these features make the Call for Life –IVR very appealing to young adults. The other advantages of IVR over SMS/text message-based intervention are that the level of literacy does not influence interaction and there are minimal chances of misunderstanding the information. The system is interoperable, can be synchronized to any database, and

information is mapped and displayed on the computer dashboard and accessed by the provider whenever needed, making it generalizable.

## Conclusion

Retention and adherence are distinct interrelated health behaviors and require tools and systems focused on them. Therefore, it is necessary to target both retention and adherence to attain viral suppression thus reducing new HIV infections, addressing health disparities, and improving health outcomes in YPLHIV. This mHealth Call for life-IVR tool has demonstrated robust performance in improving ART adherence with subsequent viral suppression and retention in care among young adults in this population.

### Recommendation

We recommend the integration of mHealth technologies into routine HIV care for youth especially those initiating ART, until they are stable in clinical care with suppressed VL.

## Supporting information

**S1 Checklist. CONSORT 2010 checklist of information to include when reporting a randomised trial\*.**
(DOC)

**S1 File. Manuscript dataset.**
(XLSX)

**S2 File. Do-file for tables.**
(DO)

**S3 File. Analysis file.**
(TXT)

**S4 File.**
(PDF)

## Acknowledgments

We acknowledge the Call for Life Youth Project staff; George Eram, and Winnie Aziku. The Academy IS team; Annet Nanungi, Francis Musinguzi, the Kiryandongo Hospital ART clinic Staff, Panyadoli, Nyakadot Health Centre ART clinic staff, and Kiryandongo District Health Office.

## Author Contributions

**Conceptualization:** Agnes Bwanika Naggirinya, David B. Meya, Agnes Kiragga, Joseph Rujumba, Peter Waiswa, Rosalind Parkes-Ratanshi.

**Data curation:** Agnes Bwanika Naggirinya, Maria Sarah Nabaggala, Grace Banturaki.

**Formal analysis:** Maria Sarah Nabaggala, Grace Banturaki, Agnes Kiragga.

**Funding acquisition:** Rosalind Parkes-Ratanshi.

**Investigation:** Agnes Bwanika Naggirinya.

**Methodology:** Agnes Bwanika Naggirinya, David B. Meya, Maria Sarah Nabaggala, Agnes Kiragga, Rosalind Parkes-Ratanshi.

**Project administration:** Agnes Bwanika Naggirinya.

**Resources:** Rosalind Parkes-Ratanshi.

**Supervision:** David B. Meya, Agnes Kiragga, Joseph Rujumba, Peter Waiswa, Rosalind Parkes-Ratanshi.

**Validation:** Agnes Bwanika Naggirinya, Agnes Kiragga, Rosalind Parkes-Ratanshi.

**Visualization:** Agnes Bwanika Naggirinya, Agnes Kiragga, Rosalind Parkes-Ratanshi.

**Writing – original draft:** Agnes Bwanika Naggirinya, David B. Meya, Rosalind Parkes-Ratanshi.

**Writing – review & editing:** Agnes Bwanika Naggirinya, David B. Meya, Maria Sarah Nabaggala, Grace Banturaki, Agnes Kiragga, Joseph Rujumba, Peter Waiswa, Rosalind Parkes-Ratanshi.

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
