## [Decision Letter · Decision Letter 0]

23 Jan 2024

PONE-D-23-37288Effectiveness of Interactive Voice Response-Call for life mHealth tool on adherence to Anti-Retroviral Therapy among young people living with HIV: A randomized trial in UgandaPLOS ONE

Dear Dr. Bwanika Naggirinya,

Thank you for submitting your manuscript to PLOS ONE. After careful consideration, we feel that it has merit but does not fully meet PLOS ONE’s publication criteria as it currently stands. Therefore, we invite you to submit a revised version of the manuscript that addresses the points raised during the review process.

Editor's comments Bwanika et al

Please follow the journal formats and be concise

Be very clear with the sample size determination: You have two outcomes (viral load suppression and retention). You state the difference between arms though not showing whether on viral load or retention! How are both outcomes taken care of in the sample size estimation?

Have a clear section on randomization concealment.

Line 167/168 talks of a final model! what was in this model and how was it estimated?

Please note that the write-up should at all times compare between arms.

Why include p-values in the baseline table? what do they mean? please check on how to report the baseline table of randomized controlled trials.

We look forward to receiving your revised manuscript.

Kind regards,

Andrew Max Abaasa, Ph.D.

Academic Editor

PLOS ONE

Journal Requirements:

"I have read the journal's policy and the authors of this manuscript have the following competing interests: RPR discloses that Infectious Diseases Institute received funds from Janssen, the Pharmaceutical Companies of Johnson and Johnson for Call for Life Projects and other Project Research."

5. In the online submission form, you indicated that [The data underlying the results presented in the study are available from the corresponding author, after acceptance of the article, since this is part of her doctoral research.]. 

6. Please amend the manuscript submission data (via Edit Submission) to include author Dr. Maria Sarah Nabaggala.

7. We note that [Figure 1] in your submission contain [map/satellite] images which may be copyrighted. All PLOS content is published under the Creative Commons Attribution License (CC BY 4.0), which means that the manuscript, images, and Supporting Information files will be freely available online, and any third party is permitted to access, download, copy, distribute, and use these materials in any way, even commercially, with proper attribution. For these reasons, we cannot publish previously copyrighted maps or satellite images created using proprietary data, such as Google software (Google Maps, Street View, and Earth). For more information, see our copyright guidelines: http://journals.plos.org/plosone/s/licenses-and-copyright.

Reviewers' comments:

Reviewer's Responses to Questions

**Comments to the Author**

1. Is the manuscript technically sound, and do the data support the conclusions?

Reviewer #1: No

Reviewer #2: Yes

2. Has the statistical analysis been performed appropriately and rigorously? 

Reviewer #1: No

Reviewer #2: Yes

3. Have the authors made all data underlying the findings in their manuscript fully available?

Reviewer #1: No

Reviewer #2: Yes

4. Is the manuscript presented in an intelligible fashion and written in standard English?

Reviewer #1: No

Reviewer #2: Yes

5. Review Comments to the Author

Reviewer #1: General comments:

This manuscript reports the results for a trial of an mHealth intervention for young people living with HIV.

Overall, I believe this manuscript has some merit, but I have some concerns about the analyses, design, and reporting.

Regarding the analyses, there is no information in the statistical methods section about which statistical methods were used. This is important for determining how significance testing was done, confidence intervals were calculated, and whether anything was done to assess differences between study characteristics. The latter is very important since Tables 3 and 4 suggest there might be potential confounders, i.e., other explanations for the differences that are found. Best I can tell, no multivariable analyses have been performed and I think the manuscript would benefit from those, even if they come to the same conclusion as the current analyses.

I also feel as though not having full baseline virology data is problematic. Not having the baseline data means it is not possible to know the changes in suppression from baseline to endline. It could be that the intervention group had a greater percentage of people with suppressed viral loads at baseline and, hence, were at greater likelihood of having a suppressed viral load at endline. I'm not sure how to rectify this, but this seems like a major problem to me.

Finally, PLOS ONE does not copyedit manuscripts and, as stated in publication criterion #5 (https://journals.plos.org/plosone/s/criteria-for-publication#loc-5), can reject manuscripts without clear descriptions. There are some incomplete sentences and strange capitalization throughout. Some parts feel almost pasted in. All of this will need to be rectified and greater care taken with the exposition.

Furthermore, the introduction could be shortened. I think the first and last paragraphs are good, but the middle is too detailed. I suggest condensing this part down into a couple paragraphs by either making only the most salient points or by summarizing the points made in as few words as possible and letting the references do the describing.

Finally, I thought the subsections (e.g., "Standard of Care (Usual Care) Arm") were useful, though one is not needed for every paragraph. I would trim down the use of these somewhat and then convert the remaining to subsection headers on their own lines and in italics as opposed to how they are in the text right now.

Specific comments:

1. (line 17) I didn't understand the clause "month 6 and 12".

2. (lines 94-98) I don't think "Health" and "Hospital" should be capitalized when they are not part of a proper noun.

3. (line 108) Please describe the algorithm or program that was used to randomize the order of the envelopes.

4. (line 132) I don't believe "Intervention" should be capitalized.

5. (lines 138-139) There should be an ethics section of the manuscript that includes IRB and trial registration as well as full information on how consent and assent were collected.

6. (lines 143-144) Again, a strange sentence when describing the month 6 and month 12 visits.

7. (lines 147-152) These two sentences are both incomplete. Plus, the openepi.com reference should be included in the bibliography and, again, strange capitalization. I suggest writing these as complete sentences and not just dumping output in the manuscript.

8. (lines 158-159) Run-on sentence. I think this sentence is supposed to define ITT but it does not.

9. (lines 165, 167) Why are you using different definitions of statistically significant?

10. (line 186, table 1) Significance testing between intervention groups is generally frowned upon because a non-significant p-value does not indicate that groups are the same. For info on the topic in relation to baseline imbalance in randomized trials see Altman, https://doi.org/10.2307/2987510 and Senn, https://doi.org/10.1002/sim.4780131703. My usual recommendation is to remove the significance testing from tables like this and use standardized mean differences (SMD) to assess imbalance (see Austin, https://doi.org/10.1080/03610910902859574). Most often, I see authors using an SMD of 0.1 or 0.2 as a threshold for assessing imbalance, though this could vary by field.

11. (lines 230-239) I believe this should be in the discussion section and not the results section.

Reviewer #2: The manuscript carries an important message in good English and is a result of a well conducted RCT. However it needs to edited to bring into a good standard for publication. The authors need to adhere to standard structure of the journal and ensure that tables are standalone with adequate description. Importantly they need to tailor the introduction to relevant information only. If the edits suggested are implemented, I think the manuscript will be of a good enough standard for publication.

6. PLOS authors have the option to publish the peer review history of their article (what does this mean?). If published, this will include your full peer review and any attached files.

Reviewer #1: No

Reviewer #2: No

---

## [Author Response · Author response to Decision Letter 0]

2 Apr 2024

Dear Editor,

We thank you and the two reviewers for time taken to provide constructive comments to better the manuscript. Please find below, the point-by- point responses (bold italics in the attachment section) to comments raised and line numbers where the changes are made. We have attached a file " Response to reviewers" too. 

Re: Point by Point response to comments raised by the reviewers and editors on PONE-D-23-37288: 

Effectiveness of Interactive Voice Response-Call for life mHealth tool on adherence to Anti-Retroviral Therapy among young people living with HIV: A randomized trial in Uganda

ACADEMIC EDITOR’S COMMENTS

• Please follow the journal formats and be concise

Thanks, this guidance has been followed in the resubmission for the revised manuscript

• Be very clear with the sample size determination: You have two outcomes (viral load suppression and retention). You state the difference between arms though not showing whether on viral load or retention! How are both outcomes taken care of in the sample size estimation?

Sample size determination was based on the primary outcome, which is viral suppression at 12 months, this has been stated in the revision as per lines 146-148: (The sample size calculation was based on primary outcome using openEpi.com sample size calculation formula for randomized clinical trials)

The difference between arms have been shown for both the primary outcome and the secondary outcomes in table 2 (Viral suppression at 12 months) and table 3 (Retention in care at 12 months, and 6 months visit),

• Have a clear section on randomization concealment.

The Randomization concealment section has been made clearer, under the study procedures subsection, thanks. Lines 104-115

• Line 167/168 talks of a final model! what was in this model and how was it estimated?

This has been included under the statistical analysis section we used log-binomial regression model, lines 155-159

• Please note that the write-up should at all times compare between arms.

Thanks for the guidance, this has been put into consideration, and updated the manuscript. 

• Why include p-values in the baseline table? what do they mean? please check on how to report the baseline table of randomized controlled trials.

After the guidance from the reviewers too, we have updated the baseline characteristics table, and calculated the standardized mean difference. This error has been rectified, thanks for pointing it out. Table 1, line 199-

REVIEWER#1

Specific comments:

1. (line 17) I didn't understand the clause "month 6 and 12".

Sorry for incomplete clause, it has been updated to; “The primary outcome was viral suppression defined as HIV RNA below 1,000 copies per mL at 12 months with retention in care and viral suppression at month 6 as a secondary outcome.” Line numbers 16-18

2. (Line numbers 94-98) I don't think "Health" and "Hospital" should be capitalized when they are not part of a proper noun.

Thanks for pointing out this, it has been taken note of and removed the capitalization in the revised version.

3. (line 108) Please describe the algorithm or program that was used to randomize the order of the envelopes.

Block randomization at 1:1 ratio was done to either standard of care (usual care) or Call for life-IVR with standard of care. Computerized varying block of six randomization was done by an independent statistician who kept the treatment code list. Line numbers 103-107

4. (line 132) I don't believe "Intervention" should be capitalized. 

Apologies once again, this has been rectified.

5. ( Line numbers 138-139) There should be an ethics section of the manuscript that includes IRB and trial registration as well as full information on how consent and assent were collected.

Thanks for the guidance, the ethics section is before the results section (line numbers 172-181) and these Line numbers have been added to the ethical and approval process section

6. ( Line numbers 143-144) Again, a strange sentence when describing the month 6 and month 12 visits.

This has been rectified to “Viral load (VL) testing was performed at baseline (for those on ART for a duration of 6 months), repeat VL samples taken at month 6, and month12 study visits. Participants were seen at month 6, and 12 visits and all procedures were repeated on these visits.” Line numbers 138-140

7. ( Line numbers 147-152) These two sentences are both incomplete. Plus, the openepi.com reference should be included in the bibliography and, again, strange capitalization. I suggest writing these as complete sentences and not just dumping output in the manuscript.

This has been revised with the phrase below: “The sample size calculation was based primary outcome using openEpi.com sample size calculation formula for randomized clinical trials (Kevin M. Sullivan, A. D., Minn M. Soe ;. (2007). "Sample Size for a Cross-Sectional, Cohort, or Clinical Trial Studies." 2019, from https://www.openepi.com/SampleSize/SSCohort.htm). We used a two-sided significance level (alpha) of 0.05, and a power of 80, with a ratio of unexposed: exposed is 1. Primary end point was viral load suppression at month 12, defined as viral load count below detection (<1000 copies/ml); We estimated 79% viral suppression in the control arm based on previous results in the study area (Ministry of Health, Uganda. Uganda Population-based HIV Impact Assessment (UPHIA) 2016-2017: Final Report. Kampala, Ministry of Health, Jul 2019) , with 94% virally suppressed in the intervention arm (De Costa, A., N. Shet A Fau - Kumarasamy, P. Kumarasamy N Fau - Ashorn, B. Ashorn P Fau - Eriksson, L. Eriksson B Fau - Bogg, V. K. Bogg L Fau - Diwan and V. K. Diwan "Design of a randomized trial to evaluate the influence of mobile phone reminders on adherence to first line antiretroviral treatment in South India--the HIVIND study protocol." (1471-2288 (Electronic)). Factoring 10% loss to follow-up, the total sample size was estimated at 206 (103 for each arm). Line numbers 144-150

8. ( Line numbers 158-159) Run-on sentence. I think this sentence is supposed to define ITT but it does not.

Thanks for pointing out this, the sentence has been updated since ITT was not used in the analysis. It has been updated as below, since per protocol analysis was used. This is the updated sentence “Per protocol analysis was conducted including only the participants attending the month 12 study visit for assessment of the proportions of viral load suppression”. Line numbers165-166

9. ( Line numbers 165, 167) Why are you using different definitions of statistically significant?

I have omitted the Confidence interval part and left the alpha of <0.05. Below is the statement in the revised manuscript

“Viral load results at month 12 were analysed, proportions of those with detectable VL and those with suppressed VL were estimated in both arms; risk ratios and factors associated with viral load failure were calculated with a p-value of 0.05 taken as statistically significant.” Line numbers 158-160

10. (line 186, table 1) Significance testing between intervention groups is generally frowned upon because a non-significant p-value does not indicate that groups are the same. For info on the topic in relation to baseline imbalance in randomized trials see Altman, https://doi.org/10.2307/2987510 and Senn, https://doi.org/10.1002/sim.4780131703. My usual recommendation is to remove the significance testing from tables like this and use standardized mean differences (SMD) to assess imbalance (see Austin, https://doi.org/10.1080/03610910902859574). Most often, I see authors using an SMD of 0.1 or 0.2 as a threshold for assessing imbalance, though this could vary by field.

Thanks for the recommendation, the significance testing has been removed- and added calculated standardized mean difference.

The baseline characteristics table (1) has been revised with SMD:

9. -State what statistical test the p-values are from. 

Thanks for this, which was erroneously missed, for the primary and secondary analysis, we used the log-binomial regression analysis, and the p-values are from the model (tables 2,3,4). Pearson Chi-square test and Fischer’s Exact tests for the bivariate analysis on reasons for missing ART (Table 5) were used. This has been explained on each table.

10. In the first paragraph of the discussion also include the findings in the SOC so the comparison is clear to the reader. - Also can the authors explain why retention was higher in the present study relative to previous mHealth studies. 

The paragraph has been revised to capture info on the SOC, below is the revision “Significant viral load suppression proportions of 73.6% in the intervention arm as compared to 51.9% in the standard of care arm at twelve months, and retention rate of 88.4% retention in care for the intervention arm as compared to 74.7% in the standard of care arm”. Line numbers 270-274

Why the retention was higher in the present study relative to previous mHealth studies; “these mHealth studies used short-text messaging service which might not be appealing to the youth compared to interactive voice response intervention. Further the IVR doesn’t depend or require the beneficiary to be literate”. Line numbers 279-281

11. References require a thorough check and improvement. References in Line numbers 478, 477, 445, 443, 418, 359 etc are not standard. 

Thanks, all the references have been improved and updated

12. Please check that the manuscript length is within the word limit for articles in this journal. 

Thanks for the advice, the length has been adjusted, removed, the limitations, authors’ contribution and funding sections.

## The supporting file S1. Manuscript dataset has been provided in an excel format and is openable now##

Thanks,

Agnes Bwanika Naggirinya

---

## [Editor Report · Decision Letter 1]

23 Apr 2024

PONE-D-23-37288R1Effectiveness of Interactive Voice Response-Call for life mHealth tool on adherence to Anti-Retroviral Therapy among young people living with HIV: A randomized trial in UgandaPLOS ONE

Dear Dr. Bwanika Naggirinya,

Thank you for submitting your manuscript to PLOS ONE. After careful consideration, we feel that it has merit but does not fully meet PLOS ONE’s publication criteria as it currently stands. Therefore, we invite you to submit a revised version of the manuscript that addresses the points raised during the review process.

We look forward to receiving your revised manuscript.

Kind regards,

Andrew Max Abaasa, Ph.D.

Academic Editor

PLOS ONE

Additional Editor Comments:

You have fairly responded to the issued raised by the editor and reviewers. Further revisions include;

(1) Please go through your manuscript sentence by sentence and improve on the language

(2) Why was Kiryandongo picked as the study district? Why those particular health centres? How were they selected from the many health centres in the district?

(3) How and where was viral load (VL) testing done from? please have a section on viral loading testing.

(4) Under ethics, what care was provided to participants with unsuppressed VL?

(5) Be specific with your outcomes; in the outcomes section, you define primary and secondary outcomes. However, in the statistical analysis, other outcomes such as VL at month 6 crop up. Please state all outcomes under the outcomes section distinguishing between primary and secondary outcomes.

(6) You need a convincing justification why you choose Per Protocol over Intention to Treat analysis

(7) I know SMD had been suggested by a reviewer, however, you do not need to show statistical differences on baseline table characteristics for a randomized controlled trial. Only use this table for intended observation on randomization and decide as necessary.

(8) Table 1 footnote, *only 75 participants had baseline VL, Statistical methods: sensitivity analysis was performed adjusting for baseline VL. How were missing VL handled. Besides the table shows more numbers with baseline VL when broken down by trial arm!

(9) Conclusion; you emphasize adherence and retention and no mention of viral suppression? While adherence and VL are linked, one isn’t a direct substitute for the other.

(10) What do you recommend from your trial

---

## [Author Response · Author response to Decision Letter 1]

20 Jun 2024

Dear Editor, 23-May-2024

We thank you for the time taken to provide constructive comments to better the manuscript. Below are the point-by-point responses (bullets underlined) to the comments raised and line numbers where the changes are made. 

Re: Point-by-Point response to additional comments raised by the editor on the manuscript “ PONE-D-23-37288R1. Effectiveness of Interactive Voice Response-Call for life mHealth tool on adherence to Anti-Retroviral Therapy among young people living with HIV: A randomized trial in Uganda”

Additional Editor Comments:

You have fairly responded to the issue raised by the editor and reviewers. Further revisions include;

(1) Please go through your manuscript sentence by sentence and improve on the language

• Thanks for the advice, I have gone through the whole manuscript, sentence by sentence, and a revised version with improved language has been submitted. 

(2) Why was Kiryandongo picked as the study district? Why those particular health centers? How were they selected from the many health centers in the district?

• Prevalence of viral load suppression (VLS) varies geographically across the country, in 2018 before the study, Kiryandongo District was among those with low viral load suppression rates (UPHIA Report, 2018), among the Infectious Diseases Institute-supported districts (18), Kiryandongo had the highest volume of adolescents (845 adolescents) with low viral load suppression rate (54%). Other districts with lower rates, did not have big numbers for this population.

• Each district in Uganda has a General Hospital and referral level Health Centre IVs at the health sub-districts. Each Health Centre IV at the health sub-district supervises several Health Centre IIIs, often with maternal health services and ambulatory care. HIV care and management in Uganda is offered at the hospital level, health centers iii, and iv (https://www.health.go.ug/docs/HSSIP10.pdf). 

• We chose Panyadoli Health Centre IV because of its volume of adolescents and young adults, and Nyakadot Health Centre III due to its maternal health services with big numbers of female gender, thus choosing these sites would represent HIV Care in the district and Uganda as a whole.

(3) How and where was viral load (VL) testing done from? please have a section on viral loading testing.

• In August 2014, the Ministry of Health initiated a centralized Viral Load (VL) testing system at Central Public Health Laboratories (CPHL), to monitor the response of Anti-retroviral therapy (ART) nationally. For all facilities involved in HIV care and management in Uganda, viral load samples are transported and tested at CPHL through the National Laboratory Sample Transport system (Hub system).

• We wanted the study to be as pragmatic as possible, so the participants’ samples were drawn as per the Ministry of Health (MoH) VL testing eligibility criteria (6- monthly for adolescents and young adults), and the National Laboratory Sample Transport System transported samples to CPHL.

• A section on VL testing has been added- line numbers 137-143

(4) Under ethics, what care was provided to participants with unsuppressed VL?

• The global standard of care for unsuppressed VL after 6 months of ART, is intensive adherence counseling (IAC), which involves psychosocial counseling & support. All participants who had been on ART for 6 months and had unsuppressed VL were offered IAC by the study research assistants and the nurse counselors in the ART clinic. The participants received monthly counseling sessions for three consecutive months- line numbers 181-184

(5) Be specific with your outcomes; in the outcomes section, you define primary and secondary outcomes. However, in the statistical analysis, other outcomes such as VL at month 6 crop up. Please state all outcomes under the outcomes section distinguishing between primary and secondary outcomes.

• The primary outcome was viral suppression at 12 months, defined as viral copies below 1000/mL at month 12 visit. The secondary outcomes were retention in care defined as the proportion of participants that turned up for care at month 12 and viral suppression at month 6 visit (line numbers 140-143).

(6) You need a convincing justification why you choose Per Protocol over Intention to Treat analysis

• We chose per-protocol (PP)analysis over intention-to-treat (ITT) analysis we think the PP produces data the is useful in decision-making, it would be more logical to assess the effect of the intervention than the effect of assignment (https://www.ahajournals.org/doi/10.1161/JAHA.122.025561). However, ITT is considered standard for most randomized clinical trials.

(7) I know SMD had been suggested by a reviewer, however, you do not need to show statistical differences on baseline table characteristics for a randomized controlled trial. Only use this table for intended observation on randomization and decide as necessary.

• Thanks for the guidance, the SMD has been removed, and the baseline characteristics table has been presented without the SMD.

(8) Table 1 footnote, *only 75 participants had baseline VL, Statistical methods: sensitivity analysis was performed adjusting for baseline VL. How were missing VL handled? Besides the table shows more numbers with baseline VL when broken down by trial arm!

• We apologize for the error in numbers, those that had below detection at baseline were 46,(21 in the intervention and 25 in the standard of care arm). This has been corrected in Table-1.

 Intervention (103) Standard of Care (103) Number 206( %)

On ART≥ 6 months(n=75) 40 (37.8%) 35 (34.9%) 75 (36.4%)

Baseline VL results* (N/A if initiating ART*) n=75 

 Detected 18 (17.5%) 11 (10.7%) 29 (38.7%)

 Below detection 21 (20.4%) 25 (24.3%) 46 (61.3%)

• Regarding missing VL, at baseline, we only analyzed those with results as the majority were not eligible for VL testing as per guidelines. We just removed those with no VL results at baseline, no imputation was done.

(9) Conclusion; you emphasize adherence and retention and no mention of viral suppression? While adherence and VL are linked, one isn’t a direct substitute for the other.

• Thanks for pointing out this, the VLS has been added in the conclusion line numbers 326-331, and it now reads:

“Retention and adherence are distinct interrelated health behaviors and require tools and systems focused on them. Therefore, it is necessary to target both retention and adherence to attain viral suppression thus reducing new HIV infections, addressing health disparities, and improving health outcomes in YPLHIV. This mHealth Call for life-IVR tool has demonstrated robust performance in improving ART adherence with subsequent viral suppression and retention in care among young adults in this population”.

(10) What do you recommend from your trial

• We have added a recommendation section- line numbers 332-334

“We recommend the integration of mHealth technologies into routine HIV care for youth especially those initiating ART, until they are stable in clinical care with suppressed VL.”

Thanks,

Agnes Bwanika Naggirinya

---

## [Editor Report · Decision Letter 2]

2 Aug 2024

Effectiveness of Interactive Voice Response-Call for life mHealth tool on adherence to Anti-Retroviral Therapy among young people living with HIV: A randomized trial in Uganda

PONE-D-23-37288R2

Dear Dr. Naggirinya

We’re pleased to inform you that your manuscript has been judged scientifically suitable for publication and will be formally accepted for publication once it meets all outstanding technical requirements.

Kind regards,

Andrew Max Abaasa, Ph.D.

Academic Editor

PLOS ONE
---

## [Editor Report · Acceptance letter]

22 Aug 2024

PONE-D-23-37288R2 

PLOS ONE

Dear Dr. Bwanika Naggirinya, 

I'm pleased to inform you that your manuscript has been deemed suitable for publication in PLOS ONE. Congratulations! Your manuscript is now being handed over to our production team.

Kind regards, 

on behalf of

Dr. Andrew Max Abaasa 

Academic Editor

PLOS ONE